# Heavy Metal Levels in Milk and Serum of Dairy Cows from Different Farms Located near an Industrial Area

**DOI:** 10.3390/ani12192574

**Published:** 2022-09-27

**Authors:** Vincenzo Monteverde, Gaetano Camilleri, Francesca Arfuso, Melissa Pennisi, Laura Perillo, Gioacchino Patitò, Gianluca Gioia, Calogero Castronovo, Giuseppe Piccione

**Affiliations:** 1Istituto Zooprofilattico Sperimentale della Sicilia “A. Mirri”, Via G. Marinuzzi, 3, 90129 Palermo, Italy; 2Department of Veterinary Sciences, University of Messina, Polo Universitario dell’Annunziata, 98168 Messina, Italy

**Keywords:** heavy metals, milk, dairy cows, serum, bioaccumulation

## Abstract

**Simple Summary:**

The increase in waste in the environment due to anthropogenic activities has a strong negative effect on the health of the Earth. Heavy metals are the most important cause of environmental pollution, and they enter into the food chain with a severe negative impact on human and animal health. The presence of heavy metals in milk probably indicates the chronic exposure of animals to these elements, suggesting the usefulness of this biological matrix as an indicator of heavy metal contamination. Therefore, the evaluation of heavy metal concentrations in milk can be a useful tool to monitor the exposure to environmental pollutions and to safeguard the security of both human and animal health status and welfare.

**Abstract:**

**Background**: Heavy metals are toxic, non-biodegradable substances able to enter the food chain of cows and then transfer to their milk. This study investigated the relationship between the heavy metal concentrations in serum and milk and the environmental pollutants exposure at two different farms in Ragusa, Italy. **Methods**: To evaluate the concentrations of aluminum (Al), chromium (Cr), iron (Fe), copper (Cu), zinc (Zn), arsenic (As), cadmium (Cd), and lead (Pb), milk and serum samples were collected from 40 Friesian dairy cows from farm 1 at about 3.7 km from an industrial area (group 1) and 40 Friesian dairy cows from farm 2 at about 400 mt from a greenhouse and 6.0 km from a chemical fertilizer factory (group 2). **Results**: The concentrations of heavy metals measured in serum and milk showed no statistically significant differences between group 1 and group 2. No significant correlation between heavy metals in serum and milk in group 1 was observed. A positive correlation between Zn concentrations measured in milk and serum samples was observed in group 2 (r = 0.35, *p* = 0.03). **Conclusions**: The determination of heavy metals in serum and milk can be an important tool to detect the exposure to environmental pollutants and in monitoring the hygienic state of the environment in which milk is produced.

## 1. Introduction

Heavy metals occur in the environment both from natural sources (e.g., soil erosion, weathering of the Earth’s crust, and mining) and anthropogenic sources (e.g., industrial effluents, urban runoff, sewage discharge, and insecticides) [1]. Increased industrial and agricultural activities have led to the significant release of different wastes containing high amounts of pollutants including heavy metals. Therefore, the industrial and agricultural activities could cause an increase in the concentrations of heavy metal compounds in the air, water, and soils, as well as their passage to the tissues and milk of grazing animals [2]. Most of the metals taken up by plants and animals can then enter the food chain, such as through dairy products [3].

In the livestock system, milk and dairy products are considered major sources of nutritious foods thanks to their macro- and micronutrients, which are essential for growth and the immune functions in both animals and humans [4,5]. Several elements in milk like iron (Fe), zinc (Zn), and copper (Cu) are essential for the human body and play a crucial role in metabolism. These elements are considered as co-factors in many enzymatic reactions showing a variety of biochemical functions in the living organism. Nevertheless, the levels of these components above the sanitary recommendations may become toxic to human health [6,7,8]. Other heavy metals such as cadmium (Cd), lead (Pb), and mercury (Hg) are nonessential elements and can cause metabolic disorders with toxic effects even at very low concentrations [8]. Besides the impact on human health, the presence of heavy metals can impact the health of cows and directly or indirectly affect milk composition [9]. Therefore, the presence of these compounds in milk, in addition to being an indirect indicator of the contamination of the environment in which the dairy cows are housed, is a direct indicator of its hygienic quality [10,11]. Heavy metals are not biodegradable; they tend to accumulate in internal organs, causing hematobiochemical and pathological alterations due to their chemical properties allowing them to escape cellular control mechanisms and bind to native proteins, DNA, and nuclear proteins, inhibiting their biological activity and resulting in toxicity and the oxidative deterioration of biological macromolecules [1,12,13]. Animals are good indicators of environmental pollution from heavy metals [14]. As a matter of fact, the chronic exposure to heavy metals through several routes results in their higher accumulation in different tissues [15], including the hair [16], blood, and urine [17] of cows. Though the heavy metal values found in the different biological matrices of several animals are an indicator of the exposure to environmental pollutants, the degree of heavy metal contamination is not constant and differs depending on the exposure routes, environmental conditions, animals’ nutrition, stage of lactation, and animal breed [18,19,20].

In view of the above considerations, the aim of the present study was to evaluate the effects of anthropogenic activity on the heavy metal concentrations in milk and serum samples collected from dairy cows housed in two different farms located in Ragusa province (Sicily). Furthermore, the possible correlation between the amount of heavy metals in milk and serum samples was investigated in order to assess the usefulness of milk as a possible bioindicator of heavy metal exposure in dairy cows.

## 2. Materials and Methods

### 2.1. Animal and Experimental Design

The study was carried out on two dairy cow farms (Figure 1) located in Ragusa, Sicily, Italy (latitude 36°55′45″48 N, longitude 14°43′4″80 E, altitude 540 mt above sea level). A total of 80 Friesian dairy cattle, with 3 ± 2 years of age and body weight of 600 ± 3 kg, were enrolled in the study. The cows produced an average of 9490.58 ± 2461.39 kg of milk per lactation, with an average of 3.64% of milk fat and 3.25% of milk protein. Specifically, 40 dairy cows were selected from farm 1 (group 1) at about 3.7 km form an industrial area (36°52′30.1″ N; 14°39′28.2″ E; 518 mt above sea level), and 40 dairy cows were from farm 2 (group 2) at about 400 mt from a greenhouse and 6.0 km from a chemical fertilizer factory (37°00′41.4″ N; 14°19′33.8″ E; 173 mt above sea level). During the research, cows were selected during the first 30–40 days of lactation, taking into account the maximum mobilization of chemical elements in the body from the depot [16]. On all farms, dairy cows were housed in barns with access to a grazing area for at least 10 h a day. They were fed a balanced diet daily (fodder, hay, and silage) and water was available ad libitum. Prior to the study, each dairy cow underwent a complete clinical examination and laboratorial tests, with a complete blood count and biochemistry obtained, which were healthy. The protocol of this study was carried out in accordance with the standards recommended by the *Guide for the Care and Use of Laboratory Animals* and Directive 2010/63/EU.

### 2.2. Sampling and Laboratory Analysis

Milk samples were collected directly from the cow’s udder in the morning during full milking, deposited in 50 mL Falcon tubes (Fisher Scientific, Waltham, MA, USA) previously washed with a 10% nitric acid (HNO_3_) solution and rinsed three times with deionized water to remove all acid content. They were preserved frozen at −60 °C until processing and analysis. Blood samples were obtained from the coccygeal vein using BD vacutainer 22 G × 25 mm needles and vacutainer tubes with clot activator and were allowed to clot overnight at 4 °C before being centrifuged at 1000 g for 20 min at 2–8 °C. The sampling was carried out by qualified and experienced personnel, avoiding unnecessary injuries and stress to the animals. Subsequently, the blood samples were refrigerated at 3 °C until digestion and analysis. The determination of heavy metals in the milk samples was performed through a digestion procedure, according to the UNI EN 13805:2002. About 0.5 mL of the milk samples was transferred into PTFE-TFM (poly-tetrafluoroethylene-tetrafluoroethylene) vessels previously decontaminated with 3 mL of ultrapure nitric acid at 60% (*v*/*v*) and 5 mL of ultrapure water. Digestion of the samples was accomplished through a Multiwave 3000 microwave digestion system (Anton-Paar, Graz, Austria). Thereafter, the vessels were cooled inside the digester under pressure for 30–45 min, then opened under a chemical hood. The solutions for the spectrometric reading in ICP-MS were prepared in a volume of 50 mL with ultrapure water. Subsequently, the samples were filtered on 0.45-micron syringe filters to reduce as much as possible the presence of interferents during the analysis phase in ICP-MS. The determination of the mercury content in the milk samples was carried out using the direct mercury analyzer, DMA-80 (Dma 80, atomic absorption spectrophotometer, Milestone, Wesleyan University, Middletown, CT, USA), without mineralizing or pretreating the sample for a reduction to metallic mercury. The samples placed in vessels (metal or quartz) were pushed by a pneumatic arm that introduced the vessels into the furnace where the samples were first dehydrated (250 °C for 60 s) and then thermally decomposed in an oxygen current (650 °C for 150 s). The mercury was then released in the form of vapor along with the combustion gases and, by means of a continuous flow of oxygen, carried through a catalyst tube. The mercury vapors were trapped by the amalgam composed of gold elements and subsequently released for quantitative determination using atomic absorption spectroscopy with a fixed wavelength of 250 nm. The analysis times varied from 6 to 10 min so that the samples had released all the mercury. The instrument was equipped with only one accessory (mercury trap/amalgamator) and consisted of a gold-plated glass tube, which bound to the combustion fumes. The determination of heavy metals in the serum samples was performed by inductively coupled plasma mass spectrometry (ICP-MS). For this analysis, 2 mL of serum from each animal was collected. To each sample, 5 mL of nitric acid at 65% (Suprapur) and 1 mL of hydrochloric acid at 37% (Suprapur) were added before mineralization in the Multiwave 3000 (Anton Paar) microwave oven. The obtained solution was filled up to a volume of 50 mL with Milli-Q water and the content of heavy metals was determined using an ICP-MS series 7700× (Agilent Technologies, Santa Monica, CA, USA). DORM-4 certified reference material was used for trace elements and other constituents (National Research Council, Irvine, CA, USA). For the determination of heavy metals content in both the milk and serum samples, the analyses were conducted on the basis of calibration lines. Specifically, ICP-MS-grade multielement calibration solutions were purchased from VWR International LTD (Randon, PA, USA) and prepared at different concentration levels from 1000 mg/L. The LoD and LoQ values obtained for each analyzed element are reported in Table 1. The linearity test was carried out through eight standards (BlankCal; 0.01 µg/L; 0.05 µg/L; 0.1 µg/L; 0.2 µg/L; 0.5 µg/L; 1 µg/L; 2 µg/L; 5 µg/L; 10 µg/L; 50 µg/L) checked by the r^2^. The linearity range was acceptable for all the elements analyzed (r^2^ > 0.999).

### 2.3. Statistical Analysis

The obtained data were expressed as mean ± standard deviation (SD). Data were analyzed for normality by the Shapiro–Wilk test. Unpaired Student’s *t*-test was used to assess statistically significant differences in the studied parameters between group 1 and group 2. The correlation between the heavy metal concentrations in the milk and blood samples obtained from dairy cows enrolled in the study was assessed by Pearson’s correlation test. The *p* values < 0.05 were considered statistically significant. Statistical analysis was performed using Prism v. 5.00 (GraphPad Software, San Diego, CA, USA).

## 3. Results

Table 2 shows mean values ± SD of the heavy metal values in the milk and blood of all 80 dairy cows from the two tested farms.

The use of unpaired Student’s *t*-tests showed no statistically significant differences in heavy metal concentrations measured both in milk and serum samples between group 1 and group 2 (*p* > 0.05).

No significant differences in the studied heavy metal concentrations between the milk and serum samples were observed in group 1 and group 2 (*p* > 0.05), except for Zn which showed higher values in serum (3.09 mg/L) than milk (2.21 mg/L) in group 2 (*p* < 0.05). Moreover, a significant positive correlation (r^2^ = 0.35; *p* < 0.05) between Zn concentrations in the milk and blood samples was observed in group 2 (Figure 2).

## 4. Discussion

The presence of nonessential elements, even at low concentrations, can lead to metabolic disorders and other serious consequences but can also cause dose-related adverse effects due to the excess of essential elements [21]. It is known that the determination of metal concentrations in milk may indirectly reflect both the direct hygienic state of the milk and dairy products and the pollution of the environment in which the milk is produced [10]. The accumulation of toxic elements in the body has a toxic effect not only on cows but also on people who consume contaminated meat and milk [22]. It has been reported that high levels of Cd, Pb, and Zn have been detected in the majority of studies conducted to determine metal concentrations in dairy products. This transmission is due mostly to industrial and human activities [2,10]. With the exception of Zn, according to the results gathered in the current study, no significant difference in the heavy metal content was observed between the serum and milk samples from each group, though their values were slightly higher in serum. It has been suggested that when metals are ingested by cows in fodder, they are easily transferred to milk and urine through the blood [17,23]. Therefore, a higher concentration in blood than milk could be expected. The presence of heavy metals in the different biological matrices of cows determines the degree of the possible environmental and trophic chain contamination; therefore, the cow could be considered as a sentinel animal [17]. A positive correlation was observed between the Zn concentrations in serum and milk in group 2, probably due to the substantial application of chemical fertilizer and manure containing high levels of heavy metals [23].

Furthermore, similar values in serum and/or milk heavy metal profiles were observed between groups 1 and 2. In accordance with our results, Arianejad et al. [24] observed no difference in the content of heavy metals between traditional and industrial farms in Iran. In contrast, Zhou et al. [9] found significant differences in milk heavy metal profiles between the farm located in an unpolluted area and the farms in industrial areas. Specifically, lower contents of Pb and Cd, a higher content of As, and a similar concentration of Cr in milk samples from the cows in an unpolluted area compared to farms in industrial areas [9]. Patra et al. [25] found significantly higher average contents of Pb and Cd in milk produced by cows reared in industrial areas compared to an unpolluted area. Di Bella et al. [26] showed that heavy metal concentrations were not relevant in meat and milk samples obtained from the farm of Augusta-Melilli Priolo (SR, Sicily). Furthermore, Di Bella et al. [26] also compared heavy metal contents in seafood products compared to other categories of foodstuffs produced in Augusta-Melilli Priolo (Syracuse province, Sicily) and found different higher values of total As, Hg, and Pb. A previous study, which assessed the heavy metal content in the hair of cattle, found higher Al, Cr, Fe, Cd, and Pb concentrations and lower Zn, Cu, and As concentrations in cows’ hair from a farm near an industrial area compared to a farm 400 mt from a greenhouse and 6 km from a chemical fertilizer factory [16]. Castro-Gonzalez et al. [17] studied heavy metal concentrations in a further biological matrix of cows reared near the Popocatepetl volcano and found a significantly higher concentration of Cd in blood than in milk and urine and of Cr in blood than in urine, and a significantly lower concentration of Pb in blood than in milk and urine and of Cu in blood than in milk. Oher authors studied the bioaccumulation of heavy metals in other samples, including blood, serum, and the tails and manes of horses living nearby the industrialized area of Milazzo, and found that Cu, Zn, and Cd showed higher values in blood samples than other tested biological substrates; moreover, they also studied cobalt and Cr, which showed higher values in tails compared to other substrates; vanadium, which showed a higher concentration in serum than in whole blood; and Pb, which showed a more significant increase in whole blood and serum compared to mane and tail hair [27].

A limitation of the current study is the missing analysis of some anthropogenic sources (i.e., industrial effluents and sewage discharge). However, this survey serves as a starting point, an animal sentinel study, so that, in future investigations enlisting a larger group of animals and other farms in the area, it may be possible to make an authorization request to the competent authorities in order to carry out environmental analyses on the possible correlation between heavy metal content from anthropogenic sources and from animal biological matrices.

## 5. Conclusions

It is well-established that the heavy metal content in the blood is an indicator of recent exposure to these compounds and the daily entry of metals into an animal’s body. Moreover, the presence of heavy metals in milk probably indicates daily exposure to these elements, suggesting the usefulness of this biological matrix as an indicator of contamination due to chronic exposure. According to the results obtained in the current study, it could be hypothesized that the evaluation of heavy metal concentrations in milk can be a useful tool for monitoring the exposure to environmental pollutions and to safeguard the security of both human and animal health status and welfare.

## Figures and Tables

**Figure 1 animals-12-02574-f001:**
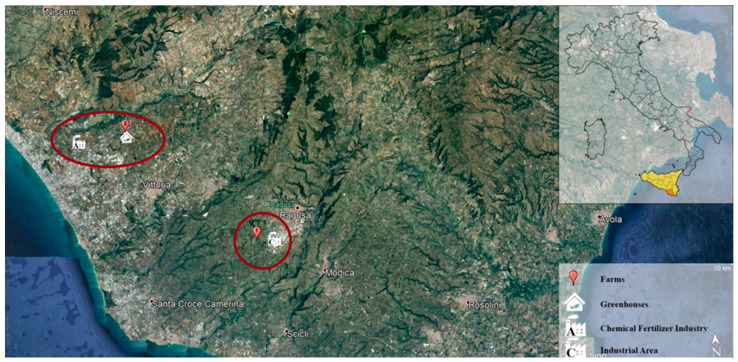
Map of the province of Ragusa, Italy, with geolocation of the two farms (farm 1, at about 3.7 km form an industrial area; farm 2, at about 400 mt from a greenhouse and 6.0 km from a chemical fertilizer factory) and anthropogenic activities.

**Figure 2 animals-12-02574-f002:**
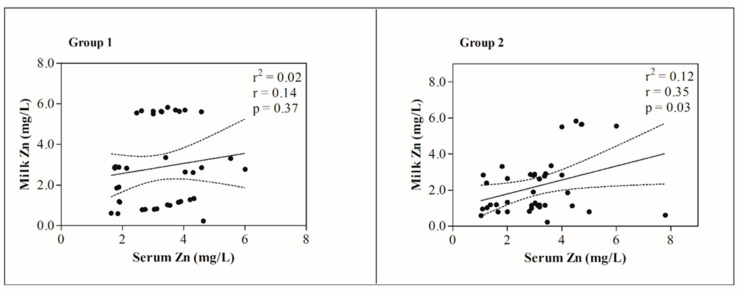
Linear regression values between the concentration of Zn measured in milk and serum samples obtained from 40 Friesian dairy cows housed in farm 1 at about 3.7 km form an industrial area (group 1) and 40 Friesian dairy cows housed in farm 2 at about 400 mt from a greenhouse and 6.0 km from a chemical fertilizer factory (group 2).

**Table 1 animals-12-02574-t001:** Method detection and quantification limits.

Element	LoD (mg/Kg)	LoQ (mg/Kg)
Cd	0.0008	0.001
Pb	0.002	0.006
Fe	0.50	1.00
Cr	0.50	1.00
Al	0.50	1.00
Cu	0.50	1.00
Zn	0.50	1.00
As	0.001	0.002

**Table 2 animals-12-02574-t002:** Mean ± standard deviation (SD) of heavy metal concentrations measured in serum and milk samples obtained from 40 Friesian dairy cows housed in farm 1 at about 3.7 km form an industrial area (group 1) and 40 Friesian dairy cows housed in farm 2 at about 400 mt from a greenhouse and 6.0 km from a chemical fertilizer factory (group 2).

Heavy Metals (mg/L)	Serum		Milk	
	*Group 1*	*Group 2*	*Group 1*	*Group 2*
Al	2.78 ± 1.90	2.76 ± 1.92	1.43 ± 0.76	1.45 ± 0.79
Cr	0.03 ± 0.02	0.03 ± 0.03	0.01 ± 0.01	0.01 ± 0.01
Fe	20.71 ± 13.90	15.48 ± 11.17	16.78 ± 2.92	14.54 ± 4.36
Cu	2.36 ± 0.64	3.06 ± 2.29	0.08 ± 0.06	0.07 ± 0.06
Zn	3.18 ± 1.12	3.09 ± 1.42 *	2.86 ± 1.93	2.21 ± 1.58
As	0.02 ± 0.01	0.01 ± 0.01	0.00 ± 0.00	0.00 ± 0.00
Cd	0.00 ± 0.00	0.00 ± 0.00	0.00 ± 0.00	0.00 ± 0.00
Pb	0.02 ± 0.01	0.03 ± 0.03	0.02 ± 0.06	0.02 ± 0.06

Significances: * vs. milk (*p* > 0.05).

## Data Availability

The data that support the findings of this study are available from the corresponding author upon reasonable request.

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
