# Peer review of "Heavy Metal Levels in Milk and Serum of Dairy Cows from Different Farms Located near an Industrial Area"

_animals, 2022, doi:10.3390/ani12192574_

Round 1
Reviewer 1 Report
Comments and suggestions for authors:
The research was done correctly. However, I don't understand why a control group was not used. The research results would then gain more.
It is important that when planning tests, cows were selected within the first 30-40 days of lactation with a fairly equal age and that each dairy cow undergoes a complete clinical and laboratory examination with complete blood count and biochemistry and a healthy result before the test. The protocol of this study was carried out in accordance with the standards recommended by Guide to the Care and Use of Laboratory Animals and Directive 99
2010/63 / UE.
My comments are purely editorial and require changes to the text:
lines 4-5 please use superscripts next to the authors' names regarding affiliation
line 85 - figure 1 is missing in the text, please complete
line177 - figure 2 is missing in the text, please complete
in Table 1, the significance of the differences in Zn content should be noted
The list of references should be harmonized as follows:
line 274 the name of the journal should be written in italics
the titles of the articles should be written as in the sentence from capital letters to lower case
lines 303-304 (reference 23) lines 313-344 (reference 26)
lines 316-317 (reference 27)
line 307 after name of journal Promo remove the dot.
Kind regards
Author Response
Reviewer 1
The research was done correctly. However, I don't understand why a control group was not used. The research results would then gain more.
It is important that when planning tests, cows were selected within the first 30-40 days of lactation with a fairly equal age and that each dairy cow undergoes a complete clinical and laboratory examination with complete blood count and biochemistry and a healthy result before the test. The protocol of this study was carried out in accordance with the standards recommended by Guide to the Care and Use of Laboratory Animals and Directive 99 2010/63 / UE.
-We sincerely thank Reviewer for the comments and suggestions which allowed us to improve the manuscript.
My comments are purely editorial and require changes to the text:
lines 4-5 please use superscripts next to the authors' names regarding affiliation
-Done
line 85 - figure 1 is missing in the text, please complete
-We thank Reviewer for pointing out this forgetfulness to us. We added the Figure 1.
line177 - figure 2 is missing in the text, please complete
-We thank Reviewer for pointing out this forgetfulness to us. We added the Figure 1.
in Table 1, the significance of the differences in Zn content should be noted
-We thank Reviewer for pointing out this forgetfulness to us. We added the significances.
The list of references should be harmonized as follows:
line 274 the name of the journal should be written in italics
-Done
the titles of the articles should be written as in the sentence from capital letters to lower case
lines 303-304 (reference 23) lines 313-344 (reference 26)
-Done
lines 316-317 (reference 27)
-Done
line 307 after name of journal Promo remove the dot.
-Done
Reviewer 2 Report
This study deals with the estimation of heavy metals in milk and blood, study contains some helpful information, however, the following suggestions must be followed to improve the draft
The introduction section is based on the harmful effects of heavy metals, this should be written on the basis of a relevant literature review of relevant studies. Only two groups were selected and milk from only 80 animals was collected. On what scientific basis were cows enrolled in this study, and what criteria were used for the selection of cows? What were the heavy metal contents of the water and feed given to cows? Groups of milk and blood should be statistically compared with each other. Milk composition should be given at the start of the discussion section. How many standards were prepared for each determination, also provided the coefficient of correlation for each estimation
Author Response
-We sincerely thank Reviewer for the comments and suggestions which allowed us to improve the manuscript. Regarding Reviewer questions: All healthy cows were enrolled in the study within the first 30-40 days of lactation with a fairly equal age. The heavy metal contents of the water and feed given to cows were not measured. We indicate that the cows produced an average of 9490.58 ± 2461.39 kg milk per lactation, with an average of 3.64% of milk-fat and 3.25% of milk-protein. We specified in the text that multielement calibration solutions were prepared at different concentration levels from 1000mg/L. The linearity test was carried out through eight standards (BlankCal; 0.01µg/L; 0.05µg/L; 0.1µg/L; 0.2µg/L; 0.5µg/L; 1µg/L; 2µg/L; 5µg/L; 10µg/L; 50µg/L) checked by the r2. The linearity range was acceptable for all the elements analyzed (r2 > 0.999). We provided Method detection and quantification limits in Table 1.
Reviewer 3 Report
COMMENTS.
This study aimed to detect heavy metals in milk and serum of dairy cows near industrial area. The findings are interesting but it does not well explained the advantages of this research and also what are the innovations about this topic. The manuscript is potentially suitable for Animals journal, but it has several flaws that requires major revisions before publication.
SIMPLE SUMMARY.
Lines 18-20. It is not clear. Please reorganize the sentence.
ABSTRACT BACKGROUND.
Line 27. Replace “metal” with “metals”.
Line 28. Replace “To evaluate concentration” with “To evaluate the concentrations”.
Lines 32-33. It is not clear. Please reorganize the sentence.
INTRODUCTION.
Lines 46-50. It is not clear and very long. Please reorganize the sentence.
MATERIALS AND METHODS.
Paragraph 2.1. Why did you not performed an analysis of some anthropogenic sources such as industrial effluents, sewage discharge, etc ?
Paragraph 2.2. Lines 126. Please remove “the” from sentence.
Paragraph 2.2. Lines 144. Add space between 50 and ml.
RESULTS.
Table 1. Please describe the results about Al and Fe, which are different between serum and milk.
DISCUSSION.
Please explain the results about Al and Fe, which are different between serum and milk.
Author Response
This study aimed to detect heavy metals in milk and serum of dairy cows near industrial area. The findings are interesting but it does not well explained the advantages of this research and also what are the innovations about this topic. The manuscript is potentially suitable for Animals journal, but it has several flaws that requires major revisions before publication.
- We sincerely thank Reviewer for the comments and suggestions which allowed us to improve the manuscript.
SIMPLE SUMMARY.
Lines 18-20. It is not clear. Please reorganize the sentence.
-We thank reviewer for his/her valuable suggestion. We improved the sentence accordingly.
ABSTRACT BACKGROUND.
Line 27. Replace “metal” with “metals”.
-Done
Line 28. Replace “To evaluate concentration” with “To evaluate the concentrations”.
-Done
Lines 32-33. It is not clear. Please reorganize the sentence.
-We thank reviewer for his/her valuable suggestion. We rewrote the sentence accordingly.
INTRODUCTION.
Lines 46-50. It is not clear and very long. Please reorganize the sentence.
-We thank reviewer for his/her valuable suggestion. We rewrote the sentence accordingly.
MATERIALS AND METHODS.
Paragraph 2.1. Why did you not performed an analysis of some anthropogenic sources such as industrial effluents, sewage discharge, etc ?
-We thank reviewer for his/her valuable question. In the current study we didn’t perform analysis of some anthropogenic sources (i.e. industrial effluents, sewage discharge, etc.) as our study wants to be a starting point, a sentinel study, so that, in future studies, we can request authorization to carry out environmental analyzes and, therefore, expand the investigation by enlisting a larger group of animals and other farms in the area.
Paragraph 2.2. Lines 126. Please remove “the” from sentence.
-Done
Paragraph 2.2. Lines 144. Add space between 5 and ml.
-Done
RESULTS.
Table 1. Please describe the results about Al and Fe, which are different between serum and milk.
-We thank Reviewer for his/her valuable comment. We checked the data and we found mistakes related to the Fe values in milk. We apologize for the error and we corrected them in table.
DISCUSSION.
Please explain the results about Al and Fe, which are different between serum and milk.
-We thank Reviewer for his/her valuable suggestion. We improved this aspect in the discussion section. We wrote “According to the results gathered in the current study, no significant difference in the heavy metal content was observed between serum and milk samples from each group, though their values were slightly higher in serum. It has been suggested that when metals are ingested by cows in fodder, they are easily transferred to milk and urine through the blood24. Presence of heavy metals in the different matrices determines the degree of environmental and trofic chain contamination with which we can consider the cow a biomarker24.”
Round 2
Reviewer 2 Report
The revised version of the manuscript may be accepted for further processing
Author Response
We sincerely thank Reviewer for the comments and suggestions of previous revision round which allowed us to improve the manuscript.
We also thank the reviewer for appreciating our review work and for accepting the revised version of the manuscript.
Reviewer 3 Report
This study has been revised but it has two important flaws that need to be addressed.
MATERIALS AND METHODS.
Paragraph 2.1. Why did you not performed an analysis of some anthropogenic sources such as industrial effluents, sewage discharge, etc ?
RESULTS.
Table 2. Please describe the results about Al and Fe, which are different between serum and milk.
DISCUSSION.
Please explain the results about Al and Fe, which are different between serum and milk.
Author Response
We sincerely thank Reviewer for the comments and suggestions which allowed us to improve the manuscript.
We thank reviewer for his/her valuable question. In the current study we didn’t perform analysis of some anthropogenic sources (i.e. industrial effluents, sewage discharge, etc.) as our study wants to be a starting point, an animal sentinel study, so that, in future studies, we can request authorization to carry out environmental analyzes for possible correlation between anthropogenic sources and animal biological matrices, and, therefore, expand the investigation by enlisting a larger group of animals and other farms in the area. We included this aspect in the manuscript as a limitation. We wrote “A limitation of the current study is the missing analysis of some anthropogenic sources (i.e. industrial effluents, sewage discharge). However, this survey wants to be a starting point, an animal sentinel study, so that, in future investigations enlisting a larger group of animals and other farms in the area, it may be possible to make the authorization request to the competent authorities in order to carry out environmental analyzes for possible correlation between heavy metals content from anthropogenic sources and from animal biological matrices.”
We checked the data and we found mistakes related to the Fe values in milk. We apologize for the error and we corrected them in table. The statistical analysis of data did not show significant difference in these heavy metals between serum and milk samples.
We improved this aspect in the discussion section. We wrote “According to the results gathered in the current study, no significant difference in the heavy metal content was observed between serum and milk samples from each group, though their values were slightly higher in serum. It has been suggested that when metals are ingested by cows in fodder, they are easily transferred to milk and urine through the blood (24). Therefore, an higher concentration in blood than milk could be expected. The presence of heavy metals in the different biological matrices from cow determines the degree of the possible environmental and trofic chain contamination, therefore, the cow could be considered as a sentinel animal (24).”
Round 3
Reviewer 3 Report
Dear Authors,
there are no comments or suggestions.